# Primary Cardiac Sarcomas: Clinical Characteristics, Management, and Outcomes at a Spanish National Reference Center

**DOI:** 10.3390/cancers17243947

**Published:** 2025-12-10

**Authors:** Carlos López-Jiménez, Mónica Benavente de Lucas, Ana Gutiérrez-Ortiz de la Tabla, Natalia Gutiérrez Alonso, Marta Arregui, Rosa Álvarez

**Affiliations:** 1Medical Oncology Department, Hospital Universitario Fundación Jiménez Díaz, 28040 Madrid, Spain; 2Instituto de Investigación Sanitaria Fundación Jiménez Diaz (IIS/FJD; UAM), 28040 Madrid, Spain; 3Medical Oncology Department, Instituto de Investigación Sanitaria Gregorio Marañón, Hospital General Universitario Gregorio Marañón, 28007 Madrid, Spain; 4Medical Oncology Department, Hospital Universitario Infanta Leonor, 28031 Madrid, Spain

**Keywords:** primary cardiac sarcoma, angiosarcoma, undifferentiated pleomorphic sarcoma, right atrium, surgery, chemotherapy, multimodal treatment, recurrence

## Abstract

Primary cardiac sarcomas are extremely rare heart tumors, and their treatment is challenging due to limited evidence. This study analyzed patients treated at a national referral center in Spain to better understand their characteristics, treatment approaches, and outcomes. Most patients initially had tumors confined to the heart, but complete surgical removal was rarely possible. Despite surgery, almost all patients experienced tumor recurrence, and many eventually developed metastases. Treatments such as chemotherapy, radiotherapy, and even heart transplantation were used, but overall survival remained low. Patients with advanced disease had particularly poor outcomes, and current therapies provided limited benefit. These findings emphasize the aggressive nature of primary cardiac sarcomas and the urgent need for more effective treatment strategies. By highlighting the patterns of disease progression and treatment response, this research aims to inform future studies and improve management approaches for this rare and deadly cancer.

## 1. Introduction

While most cardiac tumors are benign, primarily myxomas, approximately 25% are malignant, with sarcomas accounting for nearly 75% of these cases [1,2]. Primary cardiac sarcomas are extremely rare neoplasms, with an estimated incidence of 0.3 cases per 100,000 population per year and an autopsy incidence ranging from 0.001% to 0.3% [2,3]. Patients are usually diagnosed during mid-adulthood, with no clear sex predilection [1,4]. They can present with nonspecific symptoms, often leading to delayed diagnosis and advanced-stage disease at detection.

Primary cardiac sarcomas are highly heterogeneous. Angiosarcoma (AS) is the most common subtype, accounting for 20–70% of cases and predominantly involving the right heart [4,5,6,7,8,9]. Other frequent histologies include undifferentiated pleomorphic sarcoma (UPS), intimal sarcoma, and leiomyosarcoma, which frequently involve the left heart [6,8].

Metastases occur in 20–45% of patients, primarily via hematogenous spread, with the lungs being the most frequent site [4,7,10]. Central nervous system involvement occurs in 15–30% of cases and is more common than in other soft-tissue sarcomas, particularly for left-sided tumors [11].

No specific clinical guidelines exist due to the rarity of the disease, and most evidence is derived from retrospective analyses. In non-metastatic disease, complete surgical resection (R0) remains the cornerstone of therapy [5,6,12,13,14]; however, even when the primary tumor is resectable, 27–60% of patients do not survive beyond one year after surgery [5,6,13,15]. To reduce tumor burden and facilitate complete resection with negative surgical margins—often difficult due to the anatomical complexity of cardiac tumors—a multimodal approach, including perioperative chemotherapy, has been proposed by many authors [4,6,7,14,16,17,18,19], aiming to improve survival. For metastatic disease, prognosis is poor, and chemotherapy is primarily palliative, with most patients surviving less than one year [10,13,20]. Cardiac transplantation has been proposed for selected cases, although its role remains uncertain [8,21].

In this study, we aim to describe the characteristics, treatments received, and outcomes of primary cardiac sarcoma patients treated at Hospital General Universitario Gregorio Marañón in Madrid, Spain.

## 2. Materials and Methods

We performed a retrospective, observational, unicentric study. Eligible patients were those with a histologically verified diagnosis of primary cardiac sarcoma between February 2017 and December 2024.

Clinical, pathological, therapeutic, and outcome data were collected from electronic medical records.

In all cases, diagnoses were confirmed by histological review performed by a pathologist with expertise in sarcomas, including complementary studies such as immunohistochemistry (IHC), fluorescence in situ hybridization (FISH), and/or next-generation sequencing (NGS) gene panels when necessary and when the quantity and quality of the sample allowed.

Staging and response assessment were conducted using computed tomography (CT), magnetic resonance imaging (MRI), and/or positron emission tomography (PET), as appropriate. Comprehensive staging was completed for all patients at diagnosis and prior to any modification of therapy.

A descriptive analysis was performed. Recurrence-free survival (RFS) was defined as the time from surgery to local or distant relapse in patients with localized disease. Progression-free survival (PFS) was calculated from the start of each treatment line to radiological or clinical progression in patients with metastatic disease. Overall survival (OS) was defined from the date of diagnosis to death from any cause or last follow-up.

Survival curves were generated using the Kaplan–Meier method, and differences between groups were compared with the Log-rank test. *p* values < 0.05 were considered statistically significant. All analyses were performed using IBM SPSS Statistics v25 (IBM Corp., White Plains, NY, USA).

## 3. Results

### 3.1. Clinical and Pathological Characteristics

We identified 12 cases of primary cardiac sarcoma: 7 women (58.3%) and 5 men (41.7%). Median age was 43.4 years (range 13–76). Most common clinical presentations at diagnosis were dyspnea (*n* = 8, 66.7%), pericardial effusion (*n* = 6, 50%), cardiac tamponade (*n* = 4, 33.3%), thoracic pain (*n* = 4, 33.3%) and cardiac arrhythmia (*n* = 3, 25%).

Most frequent primary tumor locations were right cardiac structures (*n* = 9, 75%): right atrium (*n* = 6, 50%), pulmonary artery (*n* = 2, 16.7%) and right ventricle (*n* = 1, 8.3%). 3 tumors (25%) were located in the left atrium. Involvement of more than one cardiac structure was found in 5 patients (41.7%).

AS was the most common histologic subtype (*n* = 6, 50%), all of them located in the right atrium, followed by UPS (*n* = 4, 33.3%). One patient (8.3%) was classified as undifferentiated small round-cell sarcoma (USRCS), and in the other one (8.3%) an intimal sarcoma diagnosis was made. Regarding tumor grade, all AS were considered high-grade tumors. Three of four UPSs were classified as grade 3 (G3) and one as grade 2 (G2); the USRCS was classified as G3, and the intimal sarcoma as G2 according to the *Fédération Nationale des Centres de Lutte Contre le Cancer* (FNCLCC) classification.

A summary of the clinical and pathological features is provided in Table 1. Comprehensive per-patient histopathologic and molecular information can be found in Appendix A.

### 3.2. Management and Outcomes

7 patients (58.3%) had localized disease at diagnosis, mainly AS (*n* = 3, 42.8%) and UPS (*n* = 3, 42.8%). Surgery was performed in 6 of them (85.7%) and complete R0 resection was only achieved in 33.3% (*n* = 2). 2 patients (33.3%) received adjuvant chemotherapy (antracicline–ifosfamide combination), and cardiac transplant was performed in 1 patient (16.6%) after prior R1 surgery. All operated patients showed tumor relapse: 50% (*n* = 3) only local relapse and 50% (*n* = 3) had distant disease at first relapse diagnosis. Median RFS (mRFS) for localised disease after surgery was 5 months (CI 95% 1.5–8.6) (Figure 1). mRFS was 7.9 months (CI 95% could not be calculated due to the small sample size) for R0 (*n* = 2) vs. 4.9 months (CI 95% 0.9–8-9) for R1 patients (*n* = 4), as shown in Figure 2; and 5.0 months (CI 95% could not be calculated due to the small sample size) for adjuvant treatment (*n* = 2) vs. 4.9 months (CI 95% 0.0–11.7) for non-adjuvant treatment (*n* = 4). 6 patients with localised disease at diagnosis (85.7%) had metastatic disease at any point of disease. Median time from surgery to metastatic disease was 7.6 (CI 95% 0.0–15.8). The one patient (16.6%) on whom surgery was not performed received sequential chemotherapy (anthracycline–ifosfamide combination) and radiotherapy, with local progression 10.7 months after treatment start. Median PFS (mPFS) for localized disease at diagnosis, considering both operated and not operated patients, was 7.9 (CI 95% 0.5–15.2). Median OS (mOS) for patients with localized disease at diagnosis was 22.1 months (CI 95% 16.0–28.1) (Figure 3).

Five patients (41.7%) were diagnosed at the metastatic stage, and six of seven patients (85.7%) diagnosed at the localized stage developed distant metastases during follow-up, resulting in 11 of 12 patients (91.6%) presenting metastatic spread at some point in the course of the disease. Most frequent metastatic locations were the lungs (*n* = 6, 54.5%), brain (*n* = 2, 18.1%), liver (*n* = 2, 18.1%) and bone (*n* = 1, 9%). All metastatic patients received systemic treatment. The 1 patient (8.3%) who did not develop metastatic disease but unresectable locoregional relapse received systemic therapy as well. Of the five patients diagnosed with metastatic disease, one (20%) underwent primary tumor resection, this one showing a prolonged overall survival of 21.7 months. Median number of lines of treatment in the metastatic/unresectable disease setting was 2 (IQR 2), with 8 patients (66.6%) receiving >1 line of treatment. mPFS for the first-line treatment (*n* = 12) was 5.9 months (CI 95% 1.8–9.9) with a disease-control rate (DCR) of 66.6% (*n* = 8). mPFS was 2.1 months (CI 95% 0.0–4.4) for second-line treatment (*n* = 8).

The most commonly used drugs for advanced disease were anthracyclines (*n* = 9, 75.0%), taxanes (*n* = 7, 58.3%) and gemcitabine (*n* = 5, 41.6%), alone or in combination regimens. Anthracyclines were used as first-line treatment in 6 patients (50.0%) and second-line treatment in 3 patients (25%). Among anthracycline regimens, 5/9 (55.5%) patients received monotherapy and 4/9 (44.4%) patients a combination regimen. DCR for anthracycline-based therapy was 55.5% (*n* = 5) and mPFS for anthracycline treatment was 5.3 months (CI 95% 2.7–7.9). Among the three patients who did not receive anthracyclines for advanced disease, two had previously received the drug in the adjuvant setting. The remaining patient was initially diagnosed with kaposiform hemangioendothelioma before the definitive AS diagnosis, and anthracycline therapy was not feasible at that time.

Treatments for localized and metastatic disease are summarized in Table 2. More detailed information on specific treatments, including concrete regimens and dose ranges, is presented in Appendix A. The corresponding outcomes, including RFS, PFS, and OS, are presented in Table 3.

mOS since the diagnosis of advanced disease was 12 months (CI 95% 10.3–13.6), being similar for AS (12.0, CI 95% 2.9–21.1) vs. non-AS histology (12.0, CI 95% 9.6–14.3), *p* = 0.63. Kaplan–Meier curves for metastatic disease and comparation between AS and non-AS histology are available in Figure 4 and Figure 5.

mOS for the complete cohort, from diagnosis at any stage, was 21.7 months (95% CI 9.1–34.3), with no significant difference between AS (17.3 months, 95% CI 7.6–27.0) and non-AS histology (22.1 months, 95% CI 21.3–22.9), *p* = 0.35.

## 4. Discussion

Primary cardiac sarcomas are exceedingly rare, accounting for only a small fraction of cardiac tumors [3]. The median age at diagnosis in our cohort was 43 years, slightly younger than that reported in larger series, which usually describe onset in the fifth decade of life, and no clear sex predilection was observed. Clinical presentation in our series parallels previous reports: most patients exhibited nonspecific symptoms such as dyspnea, pericardial effusion, or chest pain, often reflecting intracardiac obstruction or heart failure rather than tumor-specific manifestations [9,22]. The right atrium was the predominant location, consistent with the known predilection of AS for right-sided cardiac chambers [4,6,7,8,23,24].

Histologically, AS was the most frequent subtype, followed by UPS, mirroring prior observations [8,23,24]. Less common entities, such as intimal sarcoma and USRCS, illustrate the histological heterogeneity of cardiac sarcomas. Given the genomic diversity of cardiac sarcomas, subtype heterogeneity may confound pooled survival analyses [19]. For example, recent molecular studies emphasize the role of MDM2 amplification in intimal sarcomas as a supportive diagnostic marker with potential prognostic relevance [8]. Disease course and sensitivity or availability of certain systemic treatments may vary depending on specific histotypes. A retrospective cohort study conducted at LMU Munich identified AS histology as statistically significantly associated with worse OS in univariate analysis [19]. Although no significant differences in OS were observed in our cohort between AS and non-AS histologies, this may be attributable to the small sample size, which represents a relevant limitation.

Nearly all tumors in our cohort were high-grade, consistent with their aggressive behavior and poor prognosis [4,24].

Surgical resection remains the cornerstone of therapy for localized disease. In our series, six of seven patients with localized tumors underwent surgery, but complete (R0) resection was achieved in only one third, reflecting the technical challenges imposed by tumor proximity to vital cardiac structures. This aligns with multicentric studies reporting R0 rates below 40%; these studies demonstrate significantly improved survival after complete resection compared with R1 or R2 resections [5,15,23]. A large population-based study including 747 patients with primary cardiac malignancies (88.5% sarcomas) showed an estimated median OS of approximately 6 months for patients who did not undergo surgery, and around 15 months for those treated surgically, finding this difference statistically significant (*p* < 0.0001) and further underscoring the central prognostic role of resection [25]. In our cohort, primary tumor resection was performed in seven patients: six with localized disease and one with metastatic disease. This likely reflects the greater opportunity for surgical intervention in the localized setting. While statistical comparison regarding primary tumor surgery in the metastatic setting was not feasible due to the small sample size, notably, the single metastatic patient who underwent surgery exhibited a prolonged OS of 21.7 months—comparable to the mOS of patients with localized disease at diagnosis (22.1 months) and exceeding that of patients with advanced disease (12 months).

Cardiac transplantation has been explored in select cases with unresectable disease [8,21]. In our cohort, one patient underwent transplantation after R1 resection, but later relapsed. Literature data remain controversial: although isolated long-term survivors exist [8], recurrence rates are high specially for AS [21]. Transplantation should therefore be reserved for highly selected cases within research protocols.

Only two patients in our series received adjuvant chemotherapy, both treated with an anthracycline–ifosfamide regimen. Retrospective analyses and systematic reviews support the use of adjuvant chemotherapy and/or radiotherapy, which can improve median OS, particularly in cases of incomplete resection, though benefits at five years remain limited [26]. Neoadjuvant chemotherapy has been proposed to facilitate resectability, extrapolating from soft tissue sarcoma experience [7,14,16,19], but evidence in cardiac sarcoma remains inconclusive due to small sample sizes and rapid tumor progression during treatment.

Radiotherapy plays a limited role because of the radiosensitivity of myocardial tissue. In our series, one patient received palliative radiotherapy, achieving transient local control. Modern conformal and proton-based modalities may offer safer delivery, but evidence remains scarce [27,28,29].

Metastatic dissemination is almost inevitable in cardiac sarcomas. In our cohort, recurrence after surgery was universal, occurring after a median RFS of 5 months—consistent with previously published data [5,7,9,15,16,19,20,24]—and resulting in a cumulative metastatic rate exceeding 90%. This is consistent with large series describing systemic progression as the rule rather than the exception [3,4,5,6,8,10,15,23,24]. The lungs were the most common site of metastasis (≈55%), followed by brain, liver, and bone, in line with prior data [24]. The relatively high incidence of brain metastases supports the inclusion of brain imaging in baseline staging, especially for left-sided or high-grade tumors [11].

All patients with metastatic disease received systemic therapy, mostly anthracycline-based regimens as the first (50%) or second (25%) line. The median PFS was 5.3 months, and DCR was 55%, comparable to that observed in advanced soft tissue sarcomas [30,31]. Subsequent lines included taxanes and gemcitabine–docetaxel combinations, with limited benefit. Weekly paclitaxel, which is particularly active in cutaneous and visceral AS, has demonstrated disease control rates of up to 70% [32], though data for cardiac AS remain anecdotal.

This study has several limitations. Firstly, those inherent to a retrospective study design. The small sample size limits the robustness of our findings and makes subgroup comparisons exploratory in nature, requiring cautious interpretation. These results are intended to be descriptive and hypothesis-generating only. Temporal bias is also relevant, since management strategies evolved during the study period (2017–2024), potentially influencing outcomes. Moreover, referral bias must be considered, as national reference centers may receive a non-representative distribution of cases. Stage-related differences are also relevant, given that nearly half of our patients presented with metastatic disease. Finally, treatment heterogeneity—including variability in resectability, surgical margins, and the use of perioperative chemotherapy or radiotherapy—may have influenced survival outcomes.

Despite these limitations, this study represents one of the largest contemporary single-center series of primary cardiac sarcomas in Spain, with all cases managed within a dedicated cardio-oncology framework, allowing for consistent diagnostic work-up, pathological review, and treatment decision-making. The uniformity of follow-up and the availability of complete clinical, surgical, and outcome data strengthen the reliability of the reported survival estimates, that align with previous large population-based studies [25] and highlighting the poor prognosis observed despite multimodal treatment strategies, with median OS in metastatic disease ranging from 6 to 12 months [7,13,16,19,23,26]. Moreover, capturing real-world management across a 7-year period provides valuable insight into current patterns of care, including the impact of multimodal treatment strategies in a national reference environment.

## 5. Conclusions

Primary cardiac sarcomas are rare and highly aggressive tumors with a poor prognosis, even after intensive multimodal treatment. Relapse after surgery with curative intent is nearly universal, and survival beyond two years is exceptional despite systemic chemotherapy.

Although the study is limited by its retrospective design, small sample size, and therapeutic heterogeneity, it provides valuable real-world evidence that reinforces the central role of surgery in localized disease and highlights the urgent need for innovative therapeutic strategies.

Future advances in managing primary cardiac sarcomas will require multicenter collaboration to standardize treatment approaches, evaluate novel therapies, and improve outcomes. Early detection, optimal surgical management, and more effective systemic treatments are essential to enhance survival in this challenging and devastating disease.

## Figures and Tables

**Figure 1 cancers-17-03947-f001:**
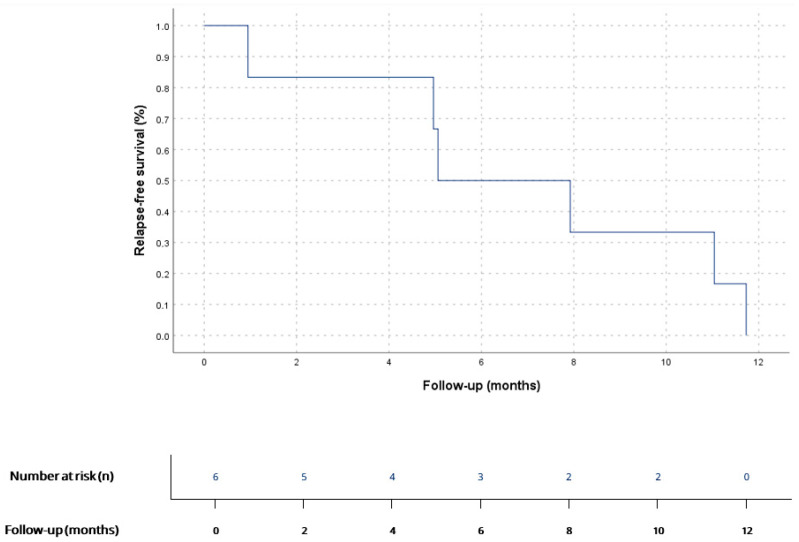
RFS for patients diagnosed with localized disease who underwent surgery (*n* = 6). According to Kaplan–Meier survival estimates, mRFS was 5 months (95% CI 1.5–8.6).

**Figure 2 cancers-17-03947-f002:**
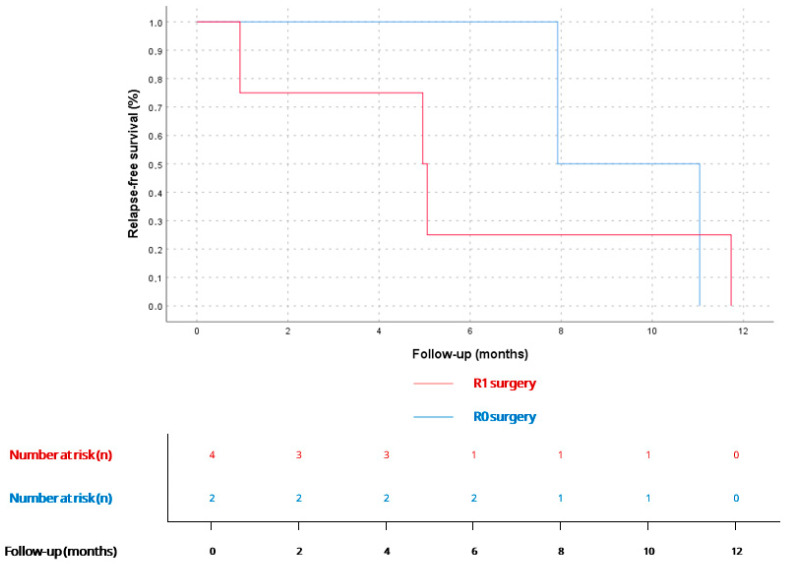
RFS for patients diagnosed with localized disease who underwent surgery (*n* = 6), stratified by R0 (*n* = 2) or R1 surgery (*n* = 4). According to Kaplan–Meier estimates, mRFS for R0 patients was 7.9 months (95% CI could not be calculated due to the small sample size), and 4.9 months (95% CI 0.9–8.9) for R1 patients (*n* = 4). No statistically significant differences were found (*p* = 0.7).

**Figure 3 cancers-17-03947-f003:**
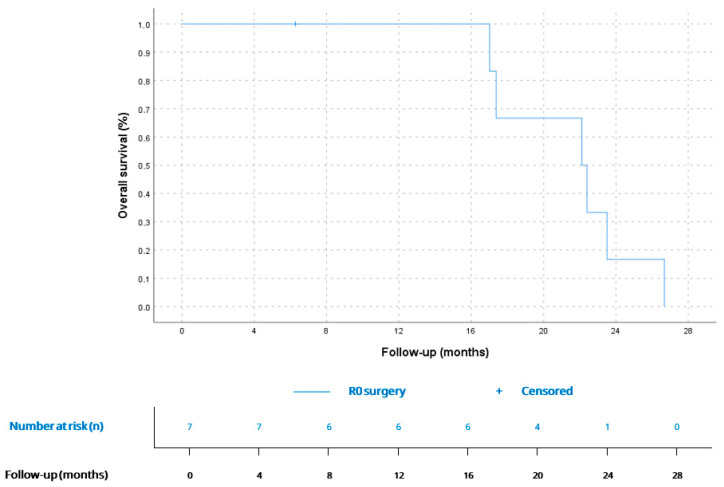
OS for patients with localized disease at diagnosis (*n* = 7). According to Kaplan–Meier survival estimates, mOS was 22.1 months (CI 95% 16.0–28.1).

**Figure 4 cancers-17-03947-f004:**
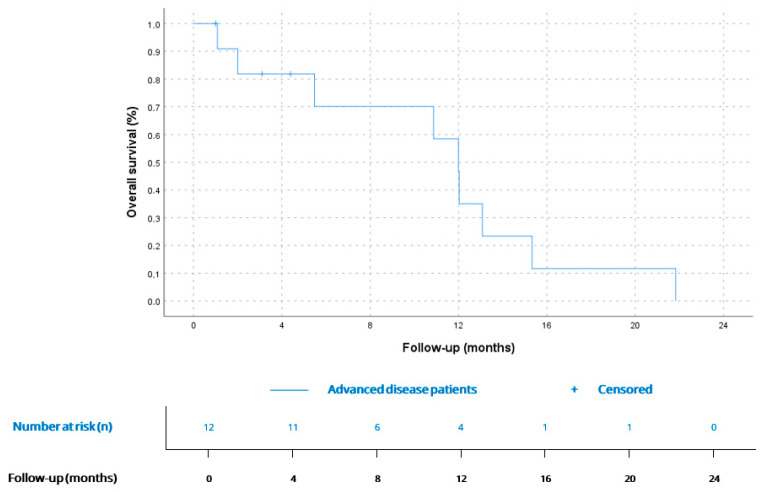
OS for patients since diagnosis of advanced disease (*n* = 12). According to Kaplan–Meier survival estimates, mOS was 12 months (CI 95% 10.3–13.6).

**Figure 5 cancers-17-03947-f005:**
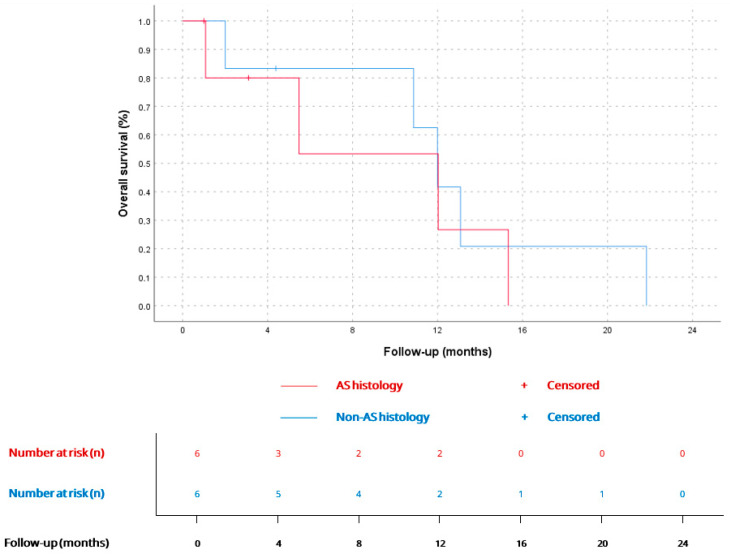
OS since advanced disease diagnosis for AS (12.0 months, CI 95% 2.9–21.1; *n* = 6) and non-AS patients (12.0 months, CI 95% 9.6–14.3; *n* = 6), *p* = 0.6.

**Table 1 cancers-17-03947-t001:** General clinical and pathological characteristics of the population.

	*n* (%)
**Gender**	
Male	5 (41.7)
Female	7 (58.3)
**Clinical presentation**	
Dyspnea	8 (66.7)
Pericardial effusion	6 (50.0)
Cardiac tamponade	4 (33.3)
Thoracic pain	4 (33.3)
Arrhythmia	3 (25.0)
**Primary tumor origin**	
Any right cardiac structure	9 (75.0)
Right atrium	6 (50.0)
Right ventricle	1 (8.3)
Pulmonary artery	2 (16.7)
Left atrium	3 (25)
Involvement of >1 cardiac chamber	5 (41.7)
**Histotype**	
Angiosarcoma	6 (50.0)
UPS	4 (33.3)
USRCS	1 (8.3)
Intimal sarcoma	1 (8.3)
**Tumor grade**	
G2	2 (16.7)
G3/High grade	10 (83.3)
**Staging at diagnosis**	
Localized disease	7 (58.3)
Metastatic disease	5 (41.7)

UPS = Undifferentiated pleomorphic sarcoma. USRCS = Undifferentiated small round cell sarcoma. G2 = Grade 2. G3 = Grade 3.

**Table 2 cancers-17-03947-t002:** Treatment details for localized and advanced disease.

		*n* (%)
**Localized disease (*n* = 7)**	**Surgery**	**6 (85.7)**
	R0	2 (33.3)
	R1	4 (66.3)
	**Adjuvant chemotherapy**	**2 (33.3)**
	**Sequential chemo-radiotherapy**	**1 (14.3)**
		***n*** **(%)**
**Advanced disease * (*n* = 12)**	Metastatic disease	11 (91.6)
	Unresectable locally advanced relapse	1 (8.33)
	Surgery	1 (8.33)
	Systemic treatment	12 (100.0)
	>1 line of treatment	8 (66.6)
	**Anthracyclines**	**9 (75.0)**
	Anthracyclines (monotherapy)	5 (55.5)
	Anthracyclines (combination regimen)	4 (44.4)
	**Taxane-based scheme**	**7 (58.3)**
	**Gemcitabine-based scheme**	**5 (41.6)**

* Both metastatic disease at diagnosis and advanced relapsing/progressing disease are considered.

**Table 3 cancers-17-03947-t003:** Main outcomes for localized and metastatic disease.

		*n*	Median (Months)	IC 95%
**Localized disease**	RFS after surgery	6	5	1.5–8.6
	Time from surgery to metastatic disease	5	7.6	0.0–15.8
	PFS	7	7.9	0.5–15.2
	OS	7	22.1	16.0–28.1
**Advanced disease**	PFS for first-line treatment	12	5.9	1.8–9.9
	PFS for second-line treatment	8	2.1	0.0–4.4
	PFS for anthracycline-based treatment	9	5.3	2.7–7.9
	OS	12	12	10.3–13.6
	OS for angiosarcoma	6	12	2.9–21.1
	OS for non-angiosarcoma histologies	6	12	9.6–14.3

RFS = Relapse-free survival. PFS = Progression-free survival. OS = Overall survival.

## Data Availability

The original contributions presented in this study are included in the article/Appendix A. Further inquiries can be directed to the corresponding author.

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
