# Peer review of "Primary Cardiac Sarcomas: Clinical Characteristics, Management, and Outcomes at a Spanish National Reference Center"

_cancers, 2025, doi:10.3390/cancers17243947_

Round 1
Reviewer 1 Report
Comments and Suggestions for Authors
The article is novelty, with significance of content, and has potential interest to readers.
The introductions provide sufficient background and include all relevant references.
The research design is appropriate.
All methods adequately described.
The results clearly presented.
The conclusions supported by the results.
Primary cardiac sarcomas are a relatively rare disease with a poor prognosis. As a rule, such patients respond to surgical treatment, often in combination with anthracycline-based chemotherapy. Anthracyclines can cause cardiotoxicity, which requires careful cardiac monitoring. Therefore, this article is clinically significant and complements other studies.
Comments on the article:
1. Remove unnecessary labels in the figures (which are present by default in SPSS).
2. For clarity, the scale in the Kaplan-Meier figures can be reduced.
Author Response
We sincerely thank the reviewer for their careful reading of our manuscript and for their valuable suggestions, which have helped us improve the clarity and quality of our work.
- Remove unnecessary labels in the figures (which are present by default in SPSS).
Response: The figures have been thoroughly reviewed to remove unnecessary labels, adjust the scales for clarity, add numbers at risk, and include 95% confidence intervals and sample sizes in the legends. - For clarity, the scale in the Kaplan-Meier figures can be reduced.
Response: The figures have been carefully revised to optimize the scale for better clarity. Numbers at risk, 95% confidence intervals, and sample sizes have been added to the legends to enhance interpretability.
We appreciate the reviewer’s efforts and constructive comments, which have substantially improved the presentation of our results.
Reviewer 2 Report
Comments and Suggestions for Authors
The current study is a retrospective, single-center observational study (2017–2024) including 12 patients with histologically confirmed primary cardiac sarcomas. The authors concluded that prognosis remains poor despite multimodal therapy, and new strategies are needed. Some major concerns must be addressed.
1- The studied sample size (n=12) is very small, as the disease is a rare one, so interpretation of the results must be modified. Authors should describe findings as descriptive or hypothesis-generating.
2- The authors must include effect sizes or survival medians with 95% Confidence Intervals in tables and avoid overinterpreting p-values.
3- The methods section must be modified. The authors must clarify how diagnoses were confirmed. Also, the imaging modalities used for staging and response assessment (CT, MRI, PET) must be stated.
4- Include the treatment in details (dose ranges and regimens for chemotherapy (anthracycline-ifosfamide, taxanes, gemcitabine). pecify median number of cycles and cumulative doses.
5- Kaplan–Meier plots (Figures 1–5) lack number-at-risk tables and confidence intervals.
6- Add legends for figures and tables explaining censoring marks and sample sizes.
7- The discussion repeats several well-known findings without highlighting what this study adds.
8- The authors can add a patient inclusion flow. Provide a diagram showing the number of patients screened, included, excluded, and followed.
9- Correct the typo errors present (i.e. anthraciclines).
Author Response
We sincerely thank the reviewer for their careful reading of our manuscript and for the constructive comments, which have greatly helped us improve the clarity and quality of our work. Our responses to each point are as follows:
- The studied sample size (n=12) is very small, as the disease is a rare one, so interpretation of the results must be modified. Authors should describe findings as descriptive or hypothesis-generating.
Response: A paragraph addressing the small sample size and acknowledging the study’s limitations has been added to the Discussion, highlighting its descriptive and hypothesis-generating purpose.
- The authors must include effect sizes or survival medians with 95% Confidence Intervals in tables and avoid overinterpreting p-values.
Response: Survival medians with 95% confidence intervals have been added where previously missing, as they were initially omitted due to small subgroup sizes. Instances where a reliable confidence interval could not be calculated due to the small sample size have been noted. A thorough check confirmed that all other values are complete.
- The methods section must be modified. The authors must clarify how diagnoses were confirmed. Also, the imaging modalities used for staging and response assessment (CT, MRI, PET) must be stated.
Response: The Methods section has been revised to clarify that all diagnoses were confirmed by histological review performed by a pathologist with expertise in sarcomas, including complementary studies when necessary. In addition, the imaging modalities used for staging and response assessment have been specified.
- Include the treatment in details (dose ranges and regimens for chemotherapy (anthracycline-ifosfamide, taxanes, gemcitabine). Specify median number of cycles and cumulative doses.
Response: Supplementary Material (Supplementary Table 2) has been provided to present details of specific treatments.
- Kaplan–Meier plots (Figures 1–5) lack number-at-risk tables and confidence intervals.
Response: Figures have been thoroughly reviewed to remove unnecessary labels, adjust scales for clarity, add numbers at risk, and include 95% confidence intervals and sample sizes in the legends.
- Add legends for figures and tables explaining censoring marks and sample sizes.
Response: Legends have been revised to include censoring marks, sample sizes, and 95% confidence intervals, improving clarity.
- The discussion repeats several well-known findings without highlighting what this study adds.
Response: The Discussion has been thoroughly revised to avoid repetition. Given the retrospective design, small sample size, and heterogeneity, the study remains descriptive, providing evidence consistent with previously reported findings in this rare disease.
- The authors can add a patient inclusion flow. Provide a diagram showing the number of patients screened, included, excluded, and followed.
Response: We thank the reviewer for the suggestion. In this study, a total of 12 patients were screened and all 12 were included in the analysis, with no exclusions. Given the small cohort size and the inclusion of all identified patients, we believe a flow diagram would not add additional clarity.
- Correct the typo errors present (i.e., anthraciclines).
Response: A thorough review to correct typographical errors has been conducted.
We greatly appreciate the reviewer’s effort and thoughtful comments, which have substantially improved the quality and clarity of our manuscript.
Reviewer 3 Report
Comments and Suggestions for Authors
The manuscript presents a retrospective analysis of 12 patients with primary cardiac sarcoma (PCS) treated at a Spanish national reference center. The study summarizes clinical characteristics, treatment patterns, and survival outcomes. Given the rarity of PCS, such institutional experience is potentially valuable. However, the current version suffers from several critical methodological, analytical, and interpretive limitations that compromise the robustness of its conclusions. The authors should carefully address the following points to improve the scientific rigor and reliability of the manuscript.
Major revisions:
- The sample size (n = 12) is critically small, rendering all subgroup analyses (e.g., R0 = 2 vs R1 = 4) statistically underpowered and prone to false-negative results. The authors should acknowledge this limitation as a primary constraint in the Discussion, replace definitive claims (e.g., “no difference”) with cautious language (“no significant difference observed”), and refrain from over-interpreting non-significant results.
- Reported median recurrence-free survival (mRFS = 7.9 months for R0 vs 4.9 months for R1) does not visually match the Kaplan–Meier curves shown in Fig. 2, where both curves converge around 5 months. According to established survival-analysis principles (see Kaplan–Meier interpretation in Andrä et al., PMID: 26156022, Fig. 3), the median should correspond to the 50% survival point on the plot. The authors must re-verify all underlying survival data to ensure textual and graphical consistency, include 95% CIs, and add risk-at-time tables beneath each plot for transparency.
- The cohort combines distinct tumor types (angiosarcoma, UPS, intimal sarcoma) without molecular verification (e.g., MDM2 amplification or EWSR1 rearrangement). Burkhard-Meier et al. ( PMID: 40571919, Fig. 1) demonstrated that cardiac sarcomas are genomically diverse and should not be analyzed as a single entity. The authors should specify molecular testing methods, provide per-patient histopathologic and molecular data in a supplementary table, and discuss how subtype heterogeneity may confound pooled survival analyses.
- The Discussion lacks comparison with major population-based analyses. Sultan et al. (PMID: 32381166, Fig. 2) evaluated 747 PCS patients in the NCDB and reported 5-year OS ≈ 5%, with surgery significantly prolonging survival. Comparing the present cohort’s OS (e.g., median ~10 months) to Sultan et al.’s Fig. 2 would situate these findings within the broader literature and reveal whether differences stem from referral bias, stage, or treatment heterogeneity. The authors should integrate such benchmarks to enhance interpretive depth.
- The Discussion acknowledges “retrospective design” but overlooks critical biases such as referral bias (more advanced cases concentrated in a national center), temporal bias (evolving treatments 2017–2024), and multimodal-therapy confounding. These should be explicitly discussed, noting how they might have inflated or underestimated survival outcomes. Clear acknowledgment of these biases will strengthen the paper’s credibility and transparency.
Minor revisions:
- Add number-at-risk tables beneath all Kaplan–Meier plots and ensure consistent numerical precision (e.g., 58 % vs 58.3 %). Clarify abbreviations (UPS, AS, RT) in table footnotes.
- The ethical review statement confirms the exemption, but should cite specific exemption clauses or legal basis to comply with international norms.
- A standardized reference format is required.
- Cancer metastasis is the major cause of cancer-related deaths and accounts for poor therapeutic outcomes. PMID:36939763 could be cited in the third paragraph of Introduction.
Author Response
We sincerely thank the reviewer for their careful and constructive review of our manuscript. The comments have greatly helped us improve the clarity, transparency, and overall quality of our work. Our responses to each point are as follows:
- The sample size (n = 12) is critically small, rendering all subgroup analyses (e.g., R0 = 2 vs R1 = 4) statistically underpowered and prone to false-negative results. The authors should acknowledge this limitation as a primary constraint in the Discussion, replace definitive claims (e.g., “no difference”) with cautious language (“no significant difference observed”), and refrain from over-interpreting non-significant results.
Response: A paragraph addressing the small sample size and acknowledging the study’s limitations has been added to the Discussion. We have also revised the language throughout the manuscript to avoid definitive claims and emphasize descriptive, hypothesis-generating interpretations.
- Reported median recurrence-free survival (mRFS = 7.9 months for R0 vs 4.9 months for R1) does not visually match the Kaplan–Meier curves shown in Fig. 2, where both curves converge around 5 months. According to established survival-analysis principles (see Kaplan–Meier interpretation in Andrä et al., PMID: 26156022, Fig. 3), the median should correspond to the 50% survival point on the plot. The authors must re-verify all underlying survival data to ensure textual and graphical consistency, include 95% CIs, and add risk-at-time tables beneath each plot for transparency.
Response: Figures have been thoroughly reviewed to remove unnecessary labels, adjust the scales for clarity, add numbers at risk, and include 95% confidence intervals and sample sizes in the legends. All survival data were cross-checked for consistency between text and figures.
- The cohort combines distinct tumor types (angiosarcoma, UPS, intimal sarcoma) without molecular verification (e.g., MDM2 amplification or EWSR1 rearrangement). Burkhard-Meier et al. (PMID: 40571919, Fig. 1) demonstrated that cardiac sarcomas are genomically diverse and should not be analyzed as a single entity. The authors should specify molecular testing methods, provide per-patient histopathologic and molecular data in a supplementary table, and discuss how subtype heterogeneity may confound pooled survival analyses.
Response: A supplementary file detailing the histological and molecular characteristics of each case has been added. The Discussion now includes consideration of how subtype heterogeneity may confound pooled survival analyses, including potential survival differences related to angiosarcoma histology.
- The Discussion lacks comparison with major population-based analyses. Sultan et al. (PMID: 32381166, Fig. 2) evaluated 747 PCS patients in the NCDB and reported 5-year OS ≈ 5%, with surgery significantly prolonging survival. Comparing the present cohort’s OS (e.g., median ~10 months) to Sultan et al.’s Fig. 2 would situate these findings within the broader literature and reveal whether differences stem from referral bias, stage, or treatment heterogeneity. The authors should integrate such benchmarks to enhance interpretive depth.
Response: The reference suggested (Sultan et al., PMID: 32381166) has been incorporated into the Discussion. A paragraph comparing our cohort’s outcomes with population-based data and discussing potential influences of referral bias, stage, and treatment heterogeneity has been added.
- The Discussion acknowledges “retrospective design” but overlooks critical biases such as referral bias (more advanced cases concentrated in a national center), temporal bias (evolving treatments 2017–2024), and multimodal-therapy confounding. These should be explicitly discussed, noting how they might have inflated or underestimated survival outcomes. Clear acknowledgment of these biases will strengthen the paper’s credibility and transparency.
Response: A paragraph explicitly addressing these potential biases has been added to the Discussion to strengthen transparency and credibility.
- Add number-at-risk tables beneath all Kaplan–Meier plots and ensure consistent numerical precision (e.g., 58 % vs 58.3 %). Clarify abbreviations (UPS, AS, RT) in table footnotes.
Response: Figures have been thoroughly reviewed to remove unnecessary labels, adjust scales, add numbers at risk, and include 95% confidence intervals and sample sizes. Abbreviations have been clarified in table footnotes, and numerical precision has been standardized.
- The ethical review statement confirms the exemption, but should cite specific exemption clauses or legal basis to comply with international norms.
Response: The paragraph has been modified to cite the specific exemption clauses and legal basis according to international norms.
- A standardized reference format is required.
Response: We have updated the manuscript to use a standardized reference format throughout, as suggested.
- Cancer metastasis is the major cause of cancer-related deaths and accounts for poor therapeutic outcomes. PMID: 36939763 could be cited in the third paragraph of Introduction.
Response: While the suggested article provides valuable insights into the molecular mechanisms of N6-methyladenosine in cancer metastasis, its focus is not directly aligned with our study, which reports on the clinical management of cardiac sarcomas at a national reference center. Therefore, we do not consider it appropriate to cite in our manuscript.
We sincerely thank the reviewer again for their effort and thoughtful comments, which have substantially improved the clarity, transparency, and quality of our manuscript.
Round 2
Reviewer 2 Report
Comments and Suggestions for Authors
The authors responded to all the comments.